

# Man-made earthquakes prevention through monitoring and discharging their causative stress-deformed states

Oleg Kuznetsov[1], Igor Chirkin[1], Ahmed Radwan[2], Ahmed Ismail[3], Yury Lyasch[4], Samuel LeRoy[5], Evgeny Rizanov[1], Sergey Koligaev[1].

[1]Department of General and Applied Geophysics, Dubna State University, Dubna, Russia
[2]Geology Department, Faculty of Science, Al-Azhar University, Assiut, Egypt
[3]Boone Pickens School of Geology, Oklahoma State University, Oklahoma, USA
[4]JYL LLC., Houston, Texas, USA
[5]Earthview Associates Inc., Houston, Texas, USA

*Correspondence to*: Ismail Ahmed (ahmed.ismail@okstate.edu)

**Abstract.** Despite our understanding of the different mechanisms of man-made earthquakes, their short-term prediction and prevention is yet to be attained. In this study, we propose an integrated four-step approach to predict and prevent man-made earthquakes or reduce their chance of occurrence. Our four-step approach includes: 1) locating the highly anomalous zones of microseismic emission (MSE) that result from the stress-deformed state inside a geological formation and often represents the "seismic nuclei" for impending earthquakes, 2) Monitoring the variations and dynamics of the anomalous MSE zones over a period of one lunar month, 3) inducing a creep-discharging of the MSE zones using a vibroseis seismic source at the ground surface, and 4) monitoring the same MSE zones following the creep-discharge to determine whether the stress-deformed state was released and the chance of potential earthquake occurrence has been eliminated or reduced. The proposed full four-step approach has never implemented at one single location. Nevertheless, the steps have been tested separately at different sites and have proven successful. We propose conducting the full four-step approach at various locations of potential man-made earthquake activities around the world including the state of Oklahoma in the United States.

## 1 Introduction:

Man-made earthquakes differ from the natural earthquakes by the fact that they occur as a result of large-scale human activities which affect the stress-deformed state of the shallow rock layers and/or the upper part of the crystalline basement over large areas. Activities that may trigger man-made earthquakes include filling of surface water reservoirs behind dams, development of oil and gas fields, extensive injection or withdrawal of fluids in the sub-surface, or injection of solids and proppant into hydrocarbon reservoirs (Hseih and Bredehoeft, 1981; Nicholson and Wesson, 1990; McGarr, et al., 2002; Lenos and Michael, 2013; Hough and Page, 2015; Frohlich et al., 2016). Such activities can provoke changes in overburden stress and pressure within the geological formations. As a result, these changes cause redistribution of both the stress-deformed state and potential elastic energy of geological formations inducing new discontinuities in the formations. A displacement of the shallow rock layers and underlying basement blocks occurs along these stress-field discontinuities. Following such rock displacement, the

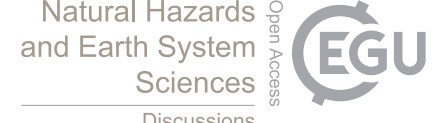



potential elastic energy of the stress-deformed state converts to kinetic energy of elastic waves causing an earthquake (Bolt, 1978, Kasahara, 1981).

Man-made earthquakes occur at relatively shallow depths of less than 7 km compared to natural earthquakes that can occur at tens of kilometers below ground surface, which can make the man-made earthquakes more destructive for a given scale of
event. McNamara (2015a, b) indicated that most induced earthquakes in Oklahoma occur in the basement at depths around 5 km. The recorded magnitude of a man-made earthquake is usually below 5.0 on the Richter scale. Nevertheless, some man-made earthquakes can result in significant destruction and loss of life, especially those that hit highly populated areas, oil and gas fields, and open-cut pits and mines.

Recently, the number of man-made earthquakes has increased dramatically and at an alarming rate in United States and Canada,
especially in regions of gas and oil shale fields containing wast disposal injection wells. According to the United States Geological Survey (USGS) and Oklahoma Geological Survey (OGS), three small earthquakes on average now occur at the state of Oklahoma every day. During the past three years, a significant increase in the number of recorded earthquakes of magnitude 3.0 and greater has been documented (Fig. 1). Because of such an alarming increase in the number of earthquakes in USA, four states have revisited hydro-fracturing operations in certain areas as an effort to reduce the increasing number of
earthquakes and resulting destruction.

In order to reduce the destruction caused by earthquakes in general, prediction of the time, location and magnitude of probable earthquakes was always a matter of research since the time of ancient civilization. Recently, we have started to see some success in long- and medium-term earthquake prediction, where researchers have delineated actively seismic zones (districts, states, regions) and made use of this information in designing earthquake-resistant construction (Karamanos, 1993).
Nevertheless, short-term prediction, which is necessary for predicting date and location of intense earthquakes in order to prepare for the consequences in time and minimize potential damage, has yet to be done successfully. In some cases, large numbers of small earthquakes at an area can be indicative of an impending large earthquake (e.g. the 1996 China major earthquake,) however, this phenomenon cannot be used for reliable short-term prediction.


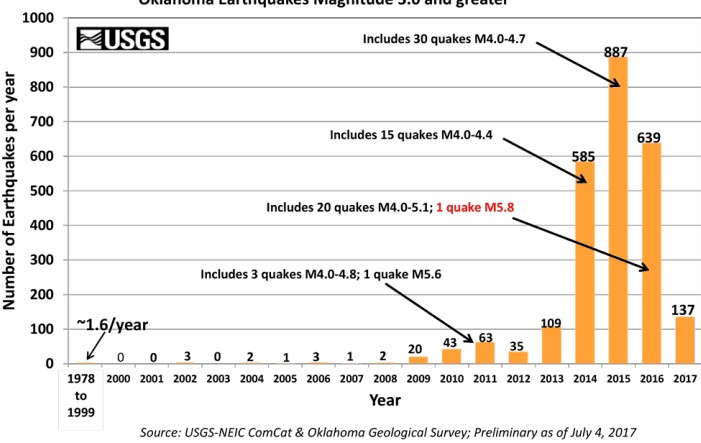


**Figure 1: Dynamics of the total number of Oklahoma earthquakes magnitude 3.0 and greater per year (USGS, 2017).**





## 1.2 The Methodology for Earthquake Prevention

The possibility of a reliable short-term prediction of earthquakes is a long-time prospect by itself because the many factors that can be responsible for triggering an earthquake and the nonlinearity of the stress accumulation and discharge within the

geological formations over time. Earthquake prediction may offer a future solution to mitigating earthquake hazards but will likely require new methodologies and innovative technologies not presently available. Considering this, we believe that the optimal solution of the earthquakes problem is to work on prevention rather than attempting non-reliable short-term prediction. The earthquake prevention we are proposing here is founded on the concept of locating the centers of the stress-deformed state inside geological formations, which could serve as a "seismic nuclei" for future earthquakes and then induce creep-discharge

of the potential energy concentrated in these centers. Most conveniently and efficiently, this proposed earthquake prevention technology will target man-made earthquakes that usually occur at shallow depths of less than 7 km. Man-made earthquake prevention would be very effective in highly populated areas of concentrated human activities, and dense infrastructure where earthquakes-caused damage will be most significant. The methodology of man-made earthquake prevention is a step-by-step surface operation that includes four steps including:

1) Locating centers (zones) of abnormally high MSE intensity by monitoring of microseismic emission (MSE) within a region of active induced seismicity,

2) Monitoring the identified abnormal MSE centers and assessing their dynamics with time. The monitoring time must go for a whole lunar month or full moon cycle. If the MSE centers grow in size and intensity, they would represent what we call "seismic nuclei" that most likely will initiate future earthquakes,

3) Creep-discharge of the MSE centers by applying surface based seismic source to get the accumulated stress to the level of the background state, and

4) Monitoring the MSE centers after the creep-discharge in order to observe the change in size and emission intensity.

A successful creep-charge occurs when the MSE anomaly is "normalized", in other words it has the microseismic emissions statistics within the MSE anomaly move towards the regional background state. This would indicate a successful reduction in

the likelihood of noticeable or damaging earthquake events arising from the targeted MSE fracture/stress concentration.

### 1.2.1 Locating centers (zones) of abnormally high Micro Seismic Emissions.

According to our methodology concept, a successful earthquake prevention will require first selecting an area of active induced seismicity and acquiring accurate microseismic monitoring surveys before and after the creep-discharge of potential energy contained in a stressed rock volume. Finding an area with induced seismicity is no longer difficult with the wide spread of

induced seismicity at many places around the world due to different human activities such as hydrofracturing and wastewater injection.





### 1.2.2 Monitoring the MSE centers

Monitoring the MSE centers before the creep-charge of them is relatively a process that requires special instrumentation, survey geometry, software and sophisticated post-acquisition processing. A geophysics team at the Scientific Petroleum Seismo-acoustic School of Russia has developed an efficient MSE monitoring technology based on what is called Seismic

Location of Emission Centers (SLEC). The SLEC method uses passive 3-D seismic scanning over selected area or targeted geological formation (Kuznetsov, et al., 2007). A schematic diagram of the SLEC technology is displayed in Figure (2). Vertically polarized geophones with relatively low central frequencies of 14 Hz are deployed in a certain array call areal antenna. The geophones can be planted on the ground surfaces or buried in shallow pits in order to attain a consistent coupling with the ground. The geophone antenna can cover an area as small as 25,000 $m^2$ and as large as 400,000 $m^2$ and each antenna

can host a number of geophones ranging between $10^2$ to $10^3$. This seismic scanning method requires continuous seismic monitoring of the seismic wave-field for a certain time window, which can range from days to weeks and the passive data are recorded as discrete samples at 10 s intervals. Once this passive seismic data are acquired, the processing will start by using the procedure of focusing the seismic scans using techniques analogous to seismic move-out procedures. This procedure allows receiving average values for the microseismic emission energy (MSE) at specified points within the volume of the geological

medium at discrete time intervals. After repeating this process for a given monitoring period at specified points within the geologic formation, we obtain an MSE time series of energy values. The concept of the radar scanning is used to locate the MSE centers (Figure 2). Because scanning is implemented in the near zone, the seismic wave field is focused on a scanned cell of the total area of the antenna by introducing time delay characteristic for each antenna sensor instead of the phased antenna array applied in the actual radar procedure. Once the energy is focused on a cell, the value of elastic energy produced

from the scanned cell within a discrete acquisition time interval can be calculated. Therefore, a time series of 500–700 hours of MSE energy produced from each scanned cell within a total period of continuous acquisition.

The SLEC method is based on the use of MSE waves that are being permanently generated everywhere in geologic formations in course of the growth and collapse of open fractures. It means that MSE waves have another kind of origin and depend on other parameters and characteristics of geologic formations compared to the conventional mirror-reflected seismic waves.

MSE waves are constantly generated in geologic formations through a growth and collapse of open fractures, driven mostly by lunar-solar solid Earth tides. SLEC technology analyzes the MSE distribution in the rock volume over time, preferably over one lunar month or a substantial portion of a month, at each imaged point within a geologic zone of interest. The MSE is a random and multiplicative statistical process. MSE observations for a given surveyed rock volume have statistical properties (such as mean energy, variance of mean energy, and autocorrelation function of MSE event) that mostly depend on the geologic

properties including stress, fracture intensity, geomechanical properties of rock formation, and type of fluid contents (oil, gas, or water) in the formation pore spaces. Reliable estimates of the geologic properties can thus be made from MSE statistical measures. (Kouznetsov, et al., 2016).



**Figure 2: Schematic ray diagram of the SLEC method implementing the focused scanning of geological formations, (Kouznetsov et al., 2016).**

Figure 3 shows the processed seismic wave field that was continuously acquired over 11-day period at 2-hour time intervals near one of the shale oil fields in the state of Texas, Kouznetsov (2016). The averaged mean energy acquired over the 11-day

15   period was presented as a 3D cube. A small size (30 x 30 x 60 m) anomalous zone stands out in the 3D cube. A nearly daily cyclic recurrence of the anomalous zone was observed owing to solid lunisolar tides. This example shows the ability of the SLEC technology to identify and monitor the MSE anomalous zones, size, intensity and dynamics.

20

25

**Figure 3: Example of 3D-distribution of MSE mean energy (within the 48-hour) estimated from SLEC acquisition (Kouznetsov et al., 2016). Example shown is from Texas (USA).**



For a better understanding of the dynamics of the anomalous MSE zones, time slices of the MSE 3D cube were generated (Fig. 4). The time slices show the dynamics of the MSE energy during periods of solid lunar tides. The variation of the MSE anomaly between the slides over 48-hours period reflects the daily variation of the causative rock fracture and stress.

### 1.2.3 Creep-discharge of Micro-Seismic Emission centers

5   An optimal method to discharge the MSE related stress centers is through using a single seismic vibrator or a group of vibrators used in the land seismic surveys. When elastic seismic waves (generated by the vibrator) propagate through the MSE center, while it has open fractures, a creep-discharge takes place at this stress center. The elastic energy partially accumulated at the ends of fractures (the stress state energy) converts to kinetic energy of elastic waves and results in a release of the stress out of the stress zone. The creep-discharge hypothesis was proved successful by the study of Kuznetsov et al., (2007). The authors

10  conducted a number of experiments during 1989-1990 to study the downhole seismo-acoustic emission using surface-based vibroseismic stimulation of oil reservoirs at a number of oil fields in Russia. Kuznetsov et al., (2007) presented a successful example of the creep-discharge technology (Fig. 5), for abnormally stressed zones at Abino-Ukranian oil field in Russia using surface-based vibroseismic stimulation of reservoirs.

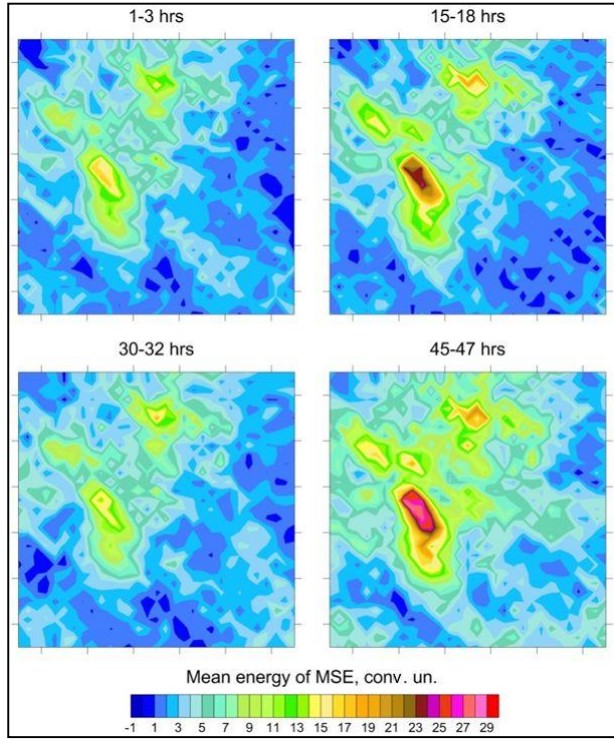

**Figure 4: Depth slice at different times within a 48-hour cycle to monitor the variation in the MSE feature showing periods of high (left side) and low (right side) lunisolar tides (Chirkin et al., 2014).**

The process started by measuring seismo-acoustic emissions (SAE) in the pay zone of one of the producing wells from the borehole logging to find out that the most intense SAE located at a depth of 2309 m (7575 feet). Seismic stimulation was



implemented from the surface (close to the wellhead) using a seismic vibrator SV-10/100 transmitting sweep signals at low-to-mid frequencies (between 12-17 Hz, 17-22 Hz, 22-27 Hz, 27-32 Hz, and 32-37 Hz) continuously for 90 seconds in each sweep. The integrated intensity of seismo-acoustic emission (SAE) was measured by a downhole noise-meter with operating within a relatively high frequency range of 100 Hz – 10 kHz continuously, starting with the first cycle of seismic stimulation. The geological formation's SAE variation corresponding to vibroseismic stimulation was measured and are displayed in Figure 5.

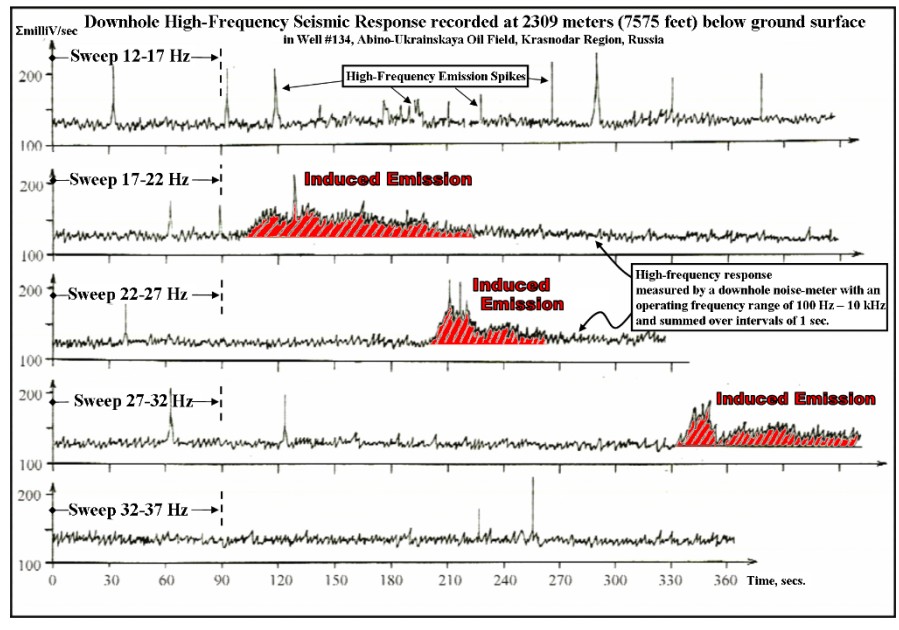

**Figure 5: Time series of the integral energy of the seismo-acoustic emission in well #134 generated from a depth interval at 2309 m below ground surface due to a surface-based vibroseismic stimulation of an oil reservoir at Abino-Ukranian field, Krasnodar region, southern Russia (Kuznetsov et al., 2007).**

One can see a selective (variable) dynamics of SAE at different stimulation frequencies. The stimulation by sweep signal of 12-17 Hz is distinguished by a somewhat increased total level of emission and short time spikes of seismic activity limited to a few seconds. At higher stimulation sweep frequency of 17-22 Hz, 22-27 Hz, and 27-32 Hz the effect of "induced emission" becomes more notable (shaded regions in Figure 5) with a duration of greater than 1 min and steep leading and gentle trailing edges. At the sweep frequency of 32-37 Hz, no "induced effect" is observed within 5 min following the stimulation start and SAE spikes are sporadic. Thus, the "induced" effect is characteristic of resonance properties of discharge of MSE centers used for the vibroseismic stimulation.

The exposure step (the time between the start of exposure and expression of response of the induced emission) increased in this case from 100 s to 330 s. This indicates that vibrational stimulation of the selected zone partially discharges the accumulated elastic energy in the form of SAE spikes and "induced" seismo-acoustic emission. Fracture "nest zones" correspond to a high fracture and stress magnitude of the rock formation. Long-term vibrational excitation resets the

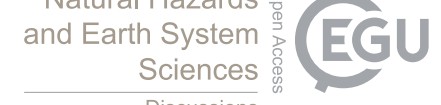

accumulated elastic energy portions in the form of bursts and induced seismic acoustic emissions. Thus, this effect presents the possibility of using vibroseis exposure to control creep-discharge centers and slowly reduce the state of stress in the rock unit.

The creep-discharge technology using the vibroseismic stimulation was also tested over an open fracture anomaly at the Staro-
5  Groznensk oilfield in the Caucasus region to prevent earthquakes (Kuznetsov, et al., 2004 and Chirkin, et al., 2014) and proved to be successful. The process was slightly different this time as it included a background (initial) survey of fracturing within the geological formations followed by vibroseismic stimulation and ended with a repeated survey for fracture detection. This fracture survey was conducted using the Side-View Seismic Location (SVSL) technology (Kuznetsov, et al., 2004 and Kuznetsov, et al., 2007). The arrangement of sources and receivers of the SVSL seismic locator arrays is shown in figure (6).
10  Background and repeated (following the stimulation) SVSL observations were made under identical conditions of emission and acquisition of seismic waves. Stimulation of the stress center for the purpose of its creep discharge was made from the surface using seismic vibrator SV-10/100. Duration of each stimulation (sweep signal) was as much as 90s and the cumulative stimulation period was about 12 hours. The total operational time was 2 days.

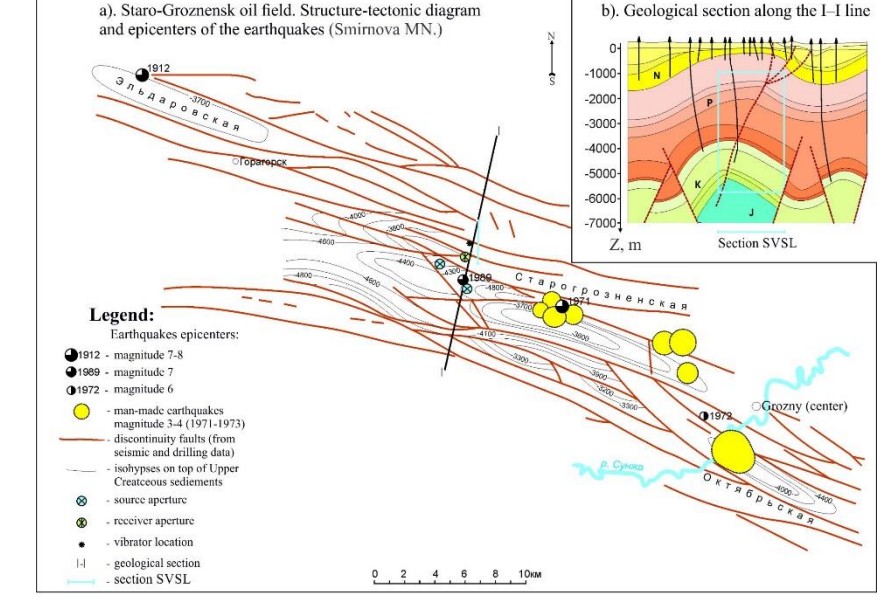

**Figure 6: a) Structure-tectonic diagram of the Staro-Groznensk oilfield site in the Caucasus region and b) the earthquake centers in**
30  **the site (Kuznetsov, et al., 2004).**

### 1.2.4 Monitoring the MSE centers after the creep-discharge

Monitoring MSE centers after the creep-discharge was conducted at the Staro-Groznensk oilfield site and compared to the before discharge data. Results of processing of SVSL data for the distributions and the intensity of open fractures at the depth interval between 1 to 6 km prior to and after the stimulation, and the difference (between before and after discharge) are

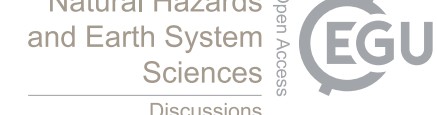

presented along the 4-km section orthogonal to a sub-vertical fault at the site (Fig. 7). The difference section demonstrates a decreased intensity of local fracture anomalies in the sub-vertical zone. This suggests a compaction in this zone and, therefore, a lower probability of occurrence of a super fracture at this site that might lead to structural block displacement. It is recommended that further analysis at this site using either the SLEC or SVSL technology would improve understanding of the

dynamics of anomalous fracture zones over longer time intervals and help determine if additional discharge of its stress state will be needed at this test location.

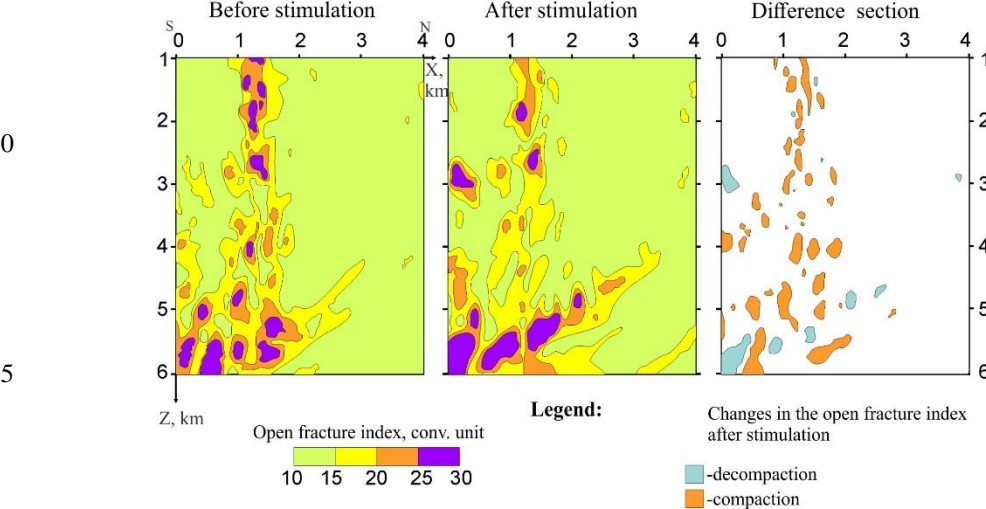

**Figure 7: Example of discharge operation applied to abnormally fractured zone of the Staro-Groznensk oilfield in the Caucasus**
**region. Depth section is based on the SVSL data acquired along the meridional line (Kuznetsov, et al., 2004)**

## 1.3 Discussion and Conclusions

The current research proposes a four-step approach to prevent or reduce the chance of occurrence of man-made earthquakes. The core concepts of this approach relies on identifying areas of anomalous MSE, determining the dynamics of these anomalous zones with respect to the lunisolar tides, creep-discharging these zones using surface seismic source followed by

monitoring the MSE zones to make sure the stress has been released. Releasing out the stress or reducing the strength of the MSE anomaly will result in preventing or reducing the chance of the occurrence of earthquakes. Despite the fact that the four steps of our earthquake prevention approach have not been jointly implemented at a single site, each individual step was tested and proved effective.

The first two steps of our approach, including locating and monitoring the anomalous MSE zones which, can be a seismic-

nuclei for a future earthquake, were successfully implemented at fractured reservoir oil fields in Texas (Kouznetsov, et al., 2016). The third step, which we consider the most innovative step of our approach, adds inducing creep-discharge of the MSE zones by using a seismic vibrosies sources at the surface. The creep-discharge approach was first tested at a borehole location at Abino-Ukranian Field, Krasnodar region in Russia (Kuznetsov et al., 2004) and proved effective.  The last step that meant



to monitor the MSE anomaly following the creep-discharge was tested at Staro-Groznensk oil field in the Caucasus region (Kuznetsov, et al., 2004).

We propose compiling the four steps at a one single location. A candidate location will be an active oil field area that is experiencing increasing induced seismicity. A place like the state of Oklahoma in the U.S. is a prime candidate for implementing this four-step approach to earthquake prevention, as Oklahoma is the most earthquake prone state in the continental US for man-made seismicity and now known as "earthquake alley". The approach needs to be repeated at several places to gain more experience and improvements in terms of data monitoring and analysis techniques.   .

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
