# Peer review of "Man-made earthquakes prevention through monitoring and discharging their causative stress-deformed states"

_Natural Hazards and Earth System Sciences, 2018_

## Referee Comment (RC1) · Anonymous Referee #1 · 18 Apr 2019

The manuscript deals with a very important issue, namely man-made/induced seismicity. The seismicity induced by human is connected with numerous activities (usually industrial) and, among others, includes: filing of water reservoirs, hydrocarbon excavation or open pit and underground mining. The issue is important because of several reasons. At least two are crucial: a) induced earthquake can be responsible for a strong ground shaking, even in case of weak or moderate quakes (because of their shallow foci), b) can be responsible for additional hazardous events like e.g. rockbursts or mining collapses. Some of them are followed by severe accidents and often terminate the industrial activity in the affected area. Taking into account all that facts, the studies focusing on prediction and prevention of man-made seismicity are of great

importance to both scientific and industrial communities.

The current version of the manuscript has numerous weak points. The most important in my opinion is that the concept was never tested in real conditions. I suggest to design such kind of studies in at least several real locations, preferable including different kind of induced seismicity. Then authors would be ready to test their 4-step approach concept and provide report on such study which would be much more suitable for a publication in NHESS.

My additional questions /suggestions are:

1) What does 'microseismic emission (MSE)' mean – please explain what kind of events belong to MSE e.g. what is the magnitude of MSE, where such events are located etc.?

2) Considering events located 5-7km below surface with small magnitudes, is it possible to obtain realistic measurements of MSE with a surface network? What are the uncertainties in such measurements?

3) In my opinion point 1 from page 3 is crucial. (i.e. "Locating centers (zones) of abnormally high MSE . . ."). As I understood (section 1.2.1), the location of these zones are simply the areas affected by induced seismicity. If so, it is probably only general selection and further very careful monitoring has to be used to identify the specific MSE zones within the aforementioned regions.

4) According to authors, the monitoring of MSE centers has to be done preferably over one lunar month. It is not clear why (maybe one solar year could be better choice) – please explain in more details.

5) The statement (page 4, 25-26): "MSE waves are constantly generated in geologic formation through a grow and collapses of open fractures, driven mostly by lunar-solar solid Earth tides" is very confusing. In my opinion the induced seismicity is rather connected with human activities and has nothing to do with lunar-solar tides - please

explain.

6) Fig 2 and 3 are very small. It is impossible to read coordinates from the axis.

7) The concept of vibration excitation by seismic vibrators may be interesting but please explain what to do in heavily populated industrial areas. In such regions these kind of field work may be very difficult to do. The same applies to areas which are located in mountain regions.

8) In my opinion conclusions are very general and not supported by any real tests. These conclusions will rather not be very useful to both scientific and industrial community.

9) References should contain more international studies. Due to lack of English written papers, it is difficult to read works which could be essential for understanding authors ideas and concepts.

---

## Author Comment (AC1) · 1 Jul 2019

Response to the reviewers' Comments.

The main remark "….The most important in my opinion is that the concept was never tested in real conditions. I suggest to design such kind of studies in at least several real locations, preferable including different kind of induced seismicity. Then authors would be ready to test their 4-step approach concept and provide report on such study which would be much more suitable for a publication in NHESS. The answer In hydrocarbon and mineral surveys conducted over the last 25 years, we have observed induced, secondary releases of seismic energy due to: 1. Stimulation by nearby micro-seismic

events recorded during passive monitoring, and 2. Stimulation of secondary events in the subsurface induced by surface sources such as vibroseismic or explosive impulses. These responses are very common in these industrial surveys and show that inground seismic stress release, these types of energy occur and also can be artificially induced from surface stimulation.

The concept of "earthquake prevention" was not tested in this study in full format under real conditions. We propose this research as as an alternative to the concept of "short-term earthquake prediction", which has been a subject of investigation for decades and only proved to be right in the case of China7.3 magnitude of the earthquake(Xicheng, 1975). Considering the urgency of solving the impact of of the earthquakes (loss of life, destruction of civil and industrial infrastructures, etc.), the scientific community needs, in our opinion, to more actively consider other options for a real solution to this problem. Therefore, the publication of the concept of "earthquake prevention" in NHESS, one of the most prestigious scientific journals of the seismological direction, will contribute to a a deep discussion aiming to develop and test this concept in a variety of real conditions. It should be noted that the main methodological and technical provisions of the concept of "earthquake prevention" have been developed and are widely used in seismic exploration in the study of oil and gas fields. First, there are modern instrumental and technical means for the formation of various patterns of active and passive observation of a seismic wave field, natural and artificially induced for any surface and deep conditions. Secondly, given that microseismic emission waves (MSE) are the main indicators of forming cracks in the geological environment, a specially developed seismic technology of the SLEC, which makes it possible to discern the energetically weak MSE waves in the observed seismic wave field (with an energy of 1-2 orders of magnitude lower than that of specularly reflected waves) and position MSE waves at their places in the rock formation. The result of the processing of seismic information using the SLEC technology is the 4D field of the MSE energy at each point of the survey of the geological environment. These results detect the spatial-temporal process of

cracking and the formation of trunk cracks in real time. Thirdly, according to the results of experimental field studies, the real possibility of using vibroseis impact for creep-discharge of hot spots in the geological environment is shown (section 4). For exposure, standard seismic vibrators can be used. The use of a group of vibrators makes it possible to form a radiating antenna, which, with the calculated radiation delays (from each vibrator), makes it possible to concentrate the energy of an artificial wave field in given zone of the geological environment. Fourthly, it can be considered that for the first time the concept of "earthquake prevention" was implemented in 1991 at the Starogroznensky oil field (Grozny, Chechnya, Russian Federation), where high seismic activity was previously observed, because of intensive water injection to increase oil production. Here, in real conditions, a vibroseismic impact was performed to discharge the foci of the stress state — zones of anomalously high open fracture, which in the future could become foci of man-made earthquakes. Unfortunately, the subsequent monitoring of fracture change (stress state) was not carried out due to the well-known events in the 90s in the Chechen Republic. Thus, it can be stated that the main elements of the "earthquake prevention" concept currently exist and are additionally created: • Instrumental and technological means of observation, included in the mode of continuous and unlimited monitoring in time; • Mathematical software for processing seismic information, included with the results of processing in real time; • Geological and tectono-physical models for interpreting the results of seismic data processing to study the 4D patterns of cracking processes and the distribution of open fractures in the geological environment in order to highlight stress centers - the "seismic nuclei" of earthquakes, and control their crypto-discharge; • Instrumental and technological means of vibroseismic impact, including the local centers of the stress state, using the focusing radiation of a group of seismic vibrators. It should be noted that these elements use MSE waves to solve a wide range of tasks in modern seismic exploration, including monitoring the process of fracturing during hydraulic fracturing and gas injection, determining the front of oil displacement of oil, vibroseismic impact on the reservoir to enhance oil recovery, etc. The obtained

results from applying these elements in various real-world conditions (when solving various tasks) suggest that the concept of "earthquake prevention" can be successfully implemented in various real-world conditions. This is clearly demonstrated in the example of the results of the work completed at the Starogroznenskoe field (section 5). According to the aforementioned, the authors believe that the postponement of the publication of this article in NHESS for testing the concept proposed by the reviewer is not a sufficiently correct proposal. Below are our answers to additional questions / suggestions from the reviewer: 1) What does 'microseismic emission (MSE)' mean – please explain what kind of events belong to MSE e.g. what is the magnitude of MSE, where such events are located etc.? Answer. The emission of elastic energy in the geological environment - the transformation of elastic energy from a potential form into kinetic. The potential form of elastic energy is the mechanical stress of rocks, which is constantly, unevenly distributed throughout the geological environment (in terms of stored energy). The kinetic form of elastic energy - elastic waves emitted by the geological medium constantly, everywhere and unevenly (in terms of radiation time, energy and spectral density of signals). The process of emission of elastic waves is a random multiplicative process at each "viewpoint", and for a volume, it is a random space-time (4D) change in the energy of elastic waves and the frequency of elastic oscillations emitted by the geological medium. The energy range of emission of elastic waves is from 10-16J to 1018J (or in magnitudes from -14.0 to 8.5), and the frequency range of elastic oscillations is from 10-1 to 107 Hz. Depending on the energy and frequency, the emission of elastic waves emitted by the geological medium is distinguished by seismic (SE) microseismic (MSE) and acoustic (AE) emissions: • SE waves occur in the earth's crust at depths from the first 1 km to $\sim$ 700 km with a magnitude of 1 to 8.5 (107 -1018 J) in the frequency range 10-1 - 102 Hz, propagate throughout the globe, form discontinuities in the geological environment , and on the surface of the Earth create earthquakes; • MSE waves can be natural and man-made genesis with a magnitude of -3.5 to 0 (1 - 106 J) in the frequency range of 100 - 103 Hz. They exist in the geological environment constantly and everywhere,

their observation, selection and positioning by modern seismic exploration it is possible at a distance (from the hypocenter) from tens of meters to ten kilometers; • AE waves occur in all rocks that are naturally occurring in the geological environment, with energy from 10-16 J (registration limit for modern equipment) to 1 J in the frequency range of 103 - 107 Hz and can be detected at a distance from 10- 2 to 102 m. There is an inverse logarithmic relationship between the value of the energy (amplitude) of the emitted wave and the period of re-radiation of the wave and the same energy range - the Gutenberg-Richter law or the law of earthquake recurrence. This law is implemented not only for the SE waves, but also for the MSE and AE, due to the fractality of the geological environment. 2) Considering events located 5-7km below surface with small magnitudes, is it possible to obtain realistic measurements of MSE with a surface network? What are the uncertainties in such measurements? Answer. To isolate seismic waves (of any type and class) in the observed seismic wave field, it is necessary that the signal-to-noise ratio (k) is $k = s / n \geq 1$, where s and n are the amplitudes of the useful wave and wave-interference. Since MSE waves are observed in the passive mode, here the main obstacle is the surface waves of natural (wind, plant vibrations, tree roots, precipitation, etc.) and man-made (operating installations, traffic, etc.) genesis. In this situation, the energy of the MSE waves is 1–2 orders of magnitude lower than the noise waves, i.e., $k = 0.1$–$0.3$. In order to isolate a useful signal, its amplitude is increased by in-phase summation of the signals of an MSE wave that has arisen at a certain place in the geomedia ("viewpoint") and has arrived at reception points (geophones) on the surface. In the case of common-mode summation, k is increased by N-0.5 times, where N is the number of receiving points. For example, the use of a 100-channel receiving antenna allows you to increase the signal-to-noise ratio by 10 times. The possibility of an unlimited increase in reception points in a ground-based observation network (a channel in a receiving antenna) is the decisive factor that allows MSE waves to be extracted with the minimum energy level necessary to solve the problem. 3) In my opinion point 1 from page 3 is crucial. (i.e. "Locating centers (zones) of abnormally high MSE . . ..."). As I understood (section

1.2.1), the location of these zones are simply the areas affected by induced seismicity. If so, it is probably only general selection and further very careful monitoring has to be used to identify the specific MSE zones within the aforementioned regions. Answer. Yes, correct. According to the results of continuous microseismic monitoring of a 4D process of cracking in a given volume of the geological environment, objects of the anomalous stress state are selected and one of them is selected for creep-discharge. Next, we perform vibration, focusing it on the selected object. Simultaneously with the impact, we continue monitoring, controlling the process of discharge of the selected object and the general redistribution of fracture in the studied volume of the medium. 4) According to authors, the monitoring of MSE centers has to be done preferably over one lunar month. It is not clear why (maybe one solar year could be better choice) – please explain in more details. Answer. The observation period, equal to the lunar month, is the initial stage of monitoring to assess the overall distribution of fracture in a given volume of the geological environment. This term of the initial stage is due to the peculiarities of the MSE process, which is random and multiplicative. The last characteristic of the process (quasi-harmonic component) determines the periodic general increase or decrease (if there is a random component) of the emission energy of elastic waves in the entire medium under study during the lunar day and month, which is associated with the appearance of the decomposition phases (the Moon in the nadir zenith) of the geological environment due to solid-state lunar ebbs and flows. Of course, the random distribution of fracture in the geological environment is also affected by other geological (tectonics, rock and reservoir pressure, gravity, etc.) and man-made factors. To eliminate the influence of the multiplicative component, it is necessary to determine the total or average value of the energy of the MSE (at each viewpoint) for the lunar month. Further, according to the results of subsequent monitoring, the change in the 3D-field of fracture is evaluated relative to the initial stage and the next primary object of vibration seismic exposure is identified. 5) The statement (page 4, 25-26): "MSE waves are constantly generated in geologic formation through a grow and collapses of open fractures, driven mostly by lunar-solar

solid Earth tides" is very confusing. In my opinion the induced seismicity is rather connected with human activities and has nothing to do with lunar-solar tides – please explain. Answer. "Caused seismicity," which is accompanied by a sharp increase in seismic activity (the number and magnitude of earthquakes), undoubtedly arises, in the overwhelming majority of cases, due to anthropogenic activity. (If we exclude the cases of volcanogenic genesis). The results of technogenic activities (reservoirs, oil and gas production, mines, open-cast mines, etc.) lead to a change in the stress state and, as a result, to the redistribution of open fracture in the geological environment. At the same time, an artificially created open fracture inherits an existing natural and, very importantly, increases the length of trunk open cracks, along which multidirectional movements of conjugate blocks occur: shifts, faults, backfills, etc. on the surface in the form of an earthquake. But the main causative factors of man-made earthquakes are artificially arisen and naturally existing open fracturing in the geological environment. In this case, artificial fracturing affects the zones of natural abnormal stress (anomalous density of open cracks), activates their growth and lengthens trunk cracks, accelerating the discharge of these anomalies, i.e. the radiation of the accumulated elastic energy. The role of lunar tides in this process is as follows. The phases of compaction and decompression of rocks created by solid-state lunar tides in the geological environment, contribute, respectively, or collapse, or growth and fusion (association) of open cracks. The process of collapse of the technogenic zone of open fracture in the phase of geological media compaction (the Moon passes at the zenith) was repeatedly observed by us during seismic monitoring of hydraulic fracturing using the SLEC technology. 6) Fig 2 and 3 are very small. It is impossible to read coordinates from the axis. Answer. Thanks for the comment, the drawings will be increased in size to be clear. 7) The concept of vibration excitation by seismic vibrators may be interesting but please explain what to do in heavily populated industrial areas. In such regions these kind of field work may be very difficult to do. The same applies to areas which are located in mountain regions. Answer. The discharge of foci of the anomalous stress state of the geological environment, located in a densely populated industrial area, is

carried out by a group of seismic vibrators, each of which is an element of a single radiating antenna. The use of an areal antenna with a size of 1-2 km in diameter allows focusing (concentrating) the energy of the vibration in any point (zone, area) of the medium where the creep-discharge object is located, which can be removed from the antenna up to several km. In this case, seismic vibrators, which will work for a long time (a month or more) and create uncomfortable conditions for the population, are located outside of their residence at a distance of 100 meters. A similar impact pattern can be used in mountainous areas, but here seismic Vibrators combined into a single antenna, are located in accessible and safe areas of mountainous terrain. 8) In my opinion conclusions are very general and not supported by any real tests. These conclusions will rather not be very useful to both scientific and industrial community. Answer. The final section (1.3) briefly summarizes the concept of "earthquake prevention", its main elements and the results of their use in real conditions of field experimental and production work, as well as during the implementation of the first positive testing of this concept at the Starogroznensky oil field. Based on the results, a conclusion was made about the feasibility of implementing the concept of "earthquake prevention" developed by the authors in Oklahoma, where in recent years there has been a sharp increase in seismicity (figure 1), which is due to man-made factors — the active development of shale oil deposits and the massive injection of hydraulic fractures proppant in the reservoir. Due to such technogenic activity, the number of seismic events in the earth has increased from $\sim$ 3 per year to $\sim$ 3 per day. The possibility of solving this problem based on the proposed concept and its practical implementation will cause, in the authors' opinion, a positive interest and will be useful not only for scientists and the business community, but, most importantly, will solve the problem of technogenic seismicity for the population of Oklahoma, and then and for other states where shale deposits are being actively developed. 9) References should contain more international studies. Due to lack of English written papers, it is difficult to read works which could be essential for understanding authors ideas and concepts. Answer. References include 12 papers, of which 8 are published in English. Thus, the

work on the subject in English in this article is present in an overwhelming number. "To understand the ideas and concepts of the authors" we recommend publication in English Kouznetsov O.L., Lyasch Y.F., Chirkin I.A., Rizanov E.G., LeRoy S.D., Koligaev S.O.: Long-term monitoring of microseismic emissions: Earth tides, fracture distribution, and fluid content // SEG, AAPG Interpretation (May 2016). 2016. V. 4, N. 2. P. T191–T204, 2016.

Please also note the supplement to this comment:
https://www.nat-hazards-earth-syst-sci-discuss.net/nhess-2018-350/nhess-2018-350-AC1-supplement.pdf

---

## Short Comment (SC1) · 18 Jul 2019

Review of the article "Man-made earthquakes prevention through monitoring and discharging their causative stress-deformed states" by Oleg Kuznetso and others. The manuscript shows a four-step approach to prevent or reduce the chance of occurrence of man-made earthquakes throughout the identification of highly anomalous zones of microseismic emission MSE due to the release of stress accumulated in the seismic dislocation zones. The four-step approach is to 1) locating the highly anomalous zones of microseismic emission (MSE), 2) Monitoring the variations and dynamics of the anomalous MSE zones over a period of one lunar month, 3) inducing a creepdischarging of the MSE zones using a vibroseis seismic source at the ground surface, and 4) monitoring the same MSE zones following the creep-discharge to determine whether the stress-deformed state was released and the chance of potential earthquake occurrence has been eliminated or reduced. I found that the article is interesting and can be considered for publication on your journal after minor revision. My only request is that, in the discussion section the authors should analyze and discuss more deeply the high amount of results presented in previous sections. One of the most important point that should be analyzed and discussed is a synthetic scenario to validate the approach.

---

## Editor Comment (EC1) · Oded Katz (Editor) · 15 Aug 2019

Dear Prof. Ismail Ahmed, Regarding your manuscript: 'Man-made earthquakes prevention through monitoring and discharging their causative stress-deformed states' (nhess-2018-350) submitted for possible publication in NHESS, I have tried to find suitable reviewers for a considerable amount of time and could only get one willing to judge your work. Thus, I decided to complete this stage and to take the editor-decision regarding your manuscript based on the single review and my own impression. Your will hear from me in the next few days. Sincerely, Oded Katz

2018-350, 2018.

---

## Author Comment (AC2) · 17 Aug 2019

The authors are deeply grateful to the distinguished reviewer Dr. Attia for the professional and attentive review of our manuscript. We certainly agree with his remark "... in the discussion section the authors should analyze and discuss more deeply the high amount of results presented in previous sections. ") In this regard, we propose adding the section below to of the discussion section. Prevention of man-made earthquakes that occur during the development of oil and gas fields is possible through: 1) obtaining and analyzing information about the 4D-field of MSE energy; 2) locating of zones with anomalously high energy of the MSE; 3) estimating the growth in size and intensity of

these zones over time; 4) identifying MSE zones prone to earthquakes; and 5) using controlled vibroseismic discharge of foci anomalously stressed state of the geomedium, which is the most likely foci of future earthquakes. Monitoring of the 4D energy field of the MSE, which reflects the process of cracking in the area is possible on the basis of a passive seismic survey using the technology "Seismic location of emission foci" [2, 3, 4]. The discharge of the foci of abnormal stress of the geological unit, identified with the "embryos" of future earthquakes, is possible using standard means of vibroseismic wave action used in seismic exploration. The change in the microcosmic emissions due to the discharge of foci SVSL technology [1, 2]. The presented examples in (Figs. 6 and 7) of the discharge of the source of the stress at Starogroznensky oil field, where earlier in 1971-73 man-made earthquakes occurred, shows the possibility of practical implementation of the idea of preventing man-made earthquakes.
* * *